# Assessment of Eotaxin Concentration in Children with Chronic Kidney Disease

**DOI:** 10.3390/ijms26157260

**Published:** 2025-07-27

**Authors:** Marta Badeńska, Andrzej Badeński, Elżbieta Świętochowska, Artur Janek, Karolina Marczak, Aleksandra Gliwińska, Maria Szczepańska

**Affiliations:** 1Department of Pediatrics, Faculty of Medical Sciences in Zabrze, Medical University of Silesia in Katowice, ul. 3 Maja 13/15, 41-800 Zabrze, Poland; marta.badenska2@gmail.com (M.B.); brylkaalex@gmail.com (A.G.); mszczepanska@sum.edu.pl (M.S.); 2Department of Medical and Molecular Biology, Faculty of Medical Sciences in Zabrze, Medical University of Silesia, 41-808 Katowice, Poland; elaswieta@interia.pl; 3Clinical Department of Pediatrics, Specialist Hospital No. 2 in Bytom, ul. Batorego 15, 41-902 Bytom, Poland; a.janek97@gmail.com; 4Department of Pediatric Nephrology with Dialysis Division for Children, Public Clinical Hospital No. 1 in Zabrze, 41-800 Zabrze, Poland; marczak.karolina9@gmail.com

**Keywords:** chronic kidney disease, eotaxin, eosinophil chemotactic factor, children

## Abstract

Chronic kidney disease (CKD) is a progressive condition which still leads to significant morbidity and mortality among patients at all ages. Its proper management should be focused on slowing down the disease sequelae, as well as establishing an early diagnosis and treatment of its complications. Eotaxin is a potent, selective eosinophil chemoattractant, which is reported to have an impact on various kidney diseases. Nevertheless, data regarding the potential correlation between eotaxin and CKD in a pediatric population is still scarce. This study aims to assess the concentration of eotaxin in children with CKD and evaluate potential correlations with selected biochemical markers and disease occurrence. Both serum and urine eotaxin concentrations were markedly higher in children with CKD compared to healthy controls. Moreover, Receiver Operating Characteristic (ROC) curves have shown that serum eotaxin and urine eotaxin levels demonstrated high sensitivity and high specificity for the allocation of patients to the study and control groups. The authors advanced a thesis that eotaxin might serve as a marker of CKD occurrence in a pediatric population. Such a research design is innovative, since it has not been analyzed in the literature yet. However, further studies are required.

## 1. Introduction

### 1.1. Chronic Kidney Disease in Pediatric Population

The diagnosis of chronic kidney disease (CKD) is established when abnormalities of kidney structure or function, accompanied by decreased estimated glomerular filtration rate (eGFR), occur and last for at least three consecutive months [1,2]. Pathogenesis of CKD in children is different than in adults and may also vary depending on age, sex and geographical location [3]. Currently, congenital anomalies of the kidneys and of the urinary tract (CAKUT) are said to be the most frequent causes of CKD in pediatric populations [4]. Moreover, focal segmental glomerulosclerosis (FSGS), polycystic kidney disease, secondary chronic glomerulonephritis, medullary cystic renal disease and hemolytic uremic syndrome (HUS) are listed among the common abnormalities triggering CKD development in children [5].

Unfortunately, the disease still leads to significant morbidity and mortality among both pediatric and adult patients [6]. Natural course of the disease as well as available therapies and recommended diet may influence vacuity and repletion in patients, resulting in nutritional alterations. Hence, CKD patients may also experience systemic effects of malnutrition, such as metabolic and endocrine abnormalities [7].

### 1.2. Eotaxin

Eotaxin, a chemokine first described in bronchoalveolar lavage of allergen-sensitized guinea pigs [8], is a potent, selective eosinophil chemoattractant. Its expression was identified in various cells, mainly in response to different inflammatory conditions including parasitic infections, asthma, atopic dermatitis, inflammatory bowel disease and to autoimmune diseases, such as rheumatoid arthritis (RA) [9,10,11]. To date, there are only several reports regarding the correlation between eotaxin levels and kidney development or function in children.

According to Starr et al. [12] eotaxin may be correlated with impaired kidney development and growth in the mouse fetus during activation of the immune system in the mother/dam. Such a research project on the youngest group of patients in a pediatric population has not yet conducted yet. Additionally, thus far, data regarding the role of eotaxin in kidney development in human fetuses is not available in the literature.

Szulimowska et al. [13] analyzed the expression of inflammatory cytokines and chemokines in saliva obtained from children with CKD. Among others, the eotaxin level was described to be elevated in CKD patients’ saliva compared to healthy controls. However, aw further assessment of the saliva profile and its role in CKD patients is required.

Chiou et al. [14] evaluated the expression of cytokines in the urothelium in children with congenital ureteropelvic junction obstruction (UPJ-O). The elevation of eotaxin was noted in urine samples and the urothelium obtained from patients with obstructed urinary tracts.

### 1.3. Aim

This study aims to assess the concentration of eotaxin in children with CKD and to evaluate potential correlations with selected biochemical markers and disease occurrence.

## 2. Results

All 73 patients fulfilled the criteria and were included in the study. Among the 32 children with CKD constituting the study group, 10 girls (31.25%) and 22 boys (68.75%) were recruited, while 41 children, including 24 boys (58.54%) and 17 girls (41.46%), represented the control group. The mean age of the entire group was 9.7 years. The multifactorial analysis aimed to assess the impact of age differentiation in the study participants. The serum and urine eotaxin concentrations in the groups of children and adolescents did not differ significantly.

The standard deviation scores (SDS) for height and weight were significantly lower in the study group. In boys, significantly higher SDS values for height and weight and significantly higher values of systolic blood pressure were observed compared to the control group. Children with CKD had significantly higher serum creatinine levels and therefore lower eGFR values compared to the control group. Higher SDS values for height and weight as well as higher eGFR values were observed in the control group. Both serum and urine eotaxin concentrations were markedly higher in the study group compared to the control group (Figure 1A,B); however, they did not differ significantly depending on the CKD stage. Considering outcomes for particular genders, urine eotaxin levels were remarkably elevated only in male patients (Table 1). The analysis also included seeking correlations between parameters indicated for both the study and control groups, such as the following: body mass index (BMI), systolic and diastolic blood pressure and mean arterial pressure (MAP); however, no significant correlations were found. Table 1 presents the characteristics of children with CKD and children from the control group. Table 2 shows the results of selected laboratory parameters obtained from children with CKD.

According to receiver operating characteristic (ROC) curves, serum eotaxin showed 88% sensitivity and 93% specificity at a cutoff point of 24.605 pg/mL, which dichotomizes the values and therefore divides patients into groups of children with CKD and healthy individuals (Figure 2).

Moreover, urine eotaxin showed 59% sensitivity and 88% specificity at a cutoff point of 10.69 pg/mL (Figure 3).

## 3. Discussion

Eotaxin is an eosinophil chemoattractant which plays a significant role in human immune responses, mainly by triggering cell movement, specifically, directed movement (chemotaxis) towards a destined location. It was widely reported in the literature and its role in kidney diseases was, and still is, being evaluated. CKD, characterized by deterioration of kidney function, can be induced by diverse pathologies, including congenital or acquired abnormalities of the immune system, which were also discussed in the literature in terms of eotaxin impact.

### 3.1. Diabetic Nephropathy

A study on patients with diabetic nephropathy (DN) [15,16], being the most frequent cause of CKD in adults, revealed a substantial involvement of eotaxin in prediction of kidney function deterioration in those cases. The authors suggested that eotaxin might be an independent predictor of the incidence of renal failure in DN patients. Another study confirmed this correlation, as Araújo et al. [17] analyzed renal biopsies from patients with DN. The aim was to assess in situ expression of cytokines and chemokines in correlation with interstitial inflammation and eGFR. A significant increase in eotaxin expression was noted in patients with DN, as was a positive correlation between higher eotaxin levels and a decrease in eGFR. Perlman et al. [18] also revealed a significant difference, although without a certain pattern, in eotaxin serum levels in patients with DN compared with healthy controls. Liu et al. [19] conducted research assessing the effectiveness of sodium-glucose cotransporter-2 (SGLT2) inhibitors in the treatment of CKD associated with diabetes mellitus type 2 (DM2). Therefore, the evaluation of markers of kidney injury, inflammation and fibrosis was performed. Eotaxin, listed among the selected molecules, reached significantly higher levels in patients treated with a SGLT2 inhibitor after 52 weeks of therapy, in comparison with the placebo. Har et al. [20] examined cytokine expression in diabetic patients with renal hyperfiltration, referred to as an equivalent of high intraglomerular pressure. Levels of eotaxin were elevated in patients with hyperfiltration. In our study, higher levels of serum and urine eotaxin were noted in children with CKD. Increased concentrations of eotaxin were previously noted in various kidney pathologies, including proliferative, acute and chronic. As mentioned, a vast majority of available studies relate to adult patients. Therefore, our research design is quite innovative regarding chronic kidney disease in children.

### 3.2. Eotaxin in Proliferative Disorders

Tao et al. [21] scrutinized the role of circulating inflammatory cytokines in the progression of renal cell carcinoma (RCC). As a result, the authors highlighted the association between higher eotaxin levels and increased RCC risk in male patients. A study related to this topic, performed by Jöhrer et al. [22], indicated that eotaxin may promote the progression of RCC, since its levels were increased in malignant tissues as opposed to a healthy kidney specimen. There was no suspicion of proliferative disorders in the studied group.

### 3.3. Eotaxin in Acute Disorders

A study performed by Andres-Hernando et al. [23] described extrarenal cytokine production and impaired renal cytokine clearance as a significant factor of increased serum cytokines in acute kidney injury (AKI). This outcome, among others, revealed increased serum eotaxin in kidney impairment (bilateral nephrectomy in an animal model) and subsequently a decrease in the molecule levels due to depletion of monocytes and phagocytes. Rabadi et al. [24], as well as Yasuda et al. [25], designed an animal model of renal ischemic injury (RII). Levels of eotaxin were elevated within hours after inducing RII. In our study, acute or chronic infections were listed as exclusion criteria; therefore, the increase in eotaxin was most likely the result of CKD.

### 3.4. Eotaxin in Chronic Disorders

Pereira et al. [26] evaluated a modulatory effect of cytokines on FSGS, using an animal model. Elevated eotaxin levels were associated with kidney failure, compared to the healthy controls. Raglianti et al. [27] reviewed clinical features and management of idiopathic retroperitoneal fibrosis (IRF). IRF is a condition that might eventually result in CKD due to obstructive uropathy. The outcome revealed increased levels of eotaxin in patients with IRF. Moreover, it was noted that higher eotaxin concentration is correlated with activation of fibroblasts and further collagen production. Research performed by Perna et al. [28] described eotaxin as a potential marker for vascular calcification (VC) in patients with CKD. VC increases risk for cardiovascular events and mortality in those patients; therefore, early diagnosis is crucial to prevent further complications. According to the study, eotaxin levels were significantly increased in patients with eGFR lower than 45 mL/min/1.73 m^2^ and in patients with VC, contrary to non-calcified individuals. Hansen et al. [29] performed a long-term observation of patients infected with West Nile Virus (WNV), who developed CKD. In the study, eotaxin, among other molecules, was selected as a factor associated with arboviral infection and kidney injury. The outcome revealed significantly increased levels of eotaxin in WNV-positive participants without CKD though. Lv et al. [30] aimed to identify potential novel predictors of heart failure (HF) in end-stage renal disease (ESRD) patients undergoing hemodialysis (HD). Among several cytokines and chemokines, eotaxin was also evaluated, showing significantly lower plasma concentrations in patients with HF in comparison with healthy controls. Mansouri et al. [31] observed higher plasma eotaxin in adult patients with CKD stage 4 and 5 compared to healthy controls. Therefore, the authors suggested the potential role of eosinophils and basophils in the pathogenesis of CKD. Pacheco-Lugo et al. [32] performed research describing the role of eotaxin in patients with lupus nephritis (LN). The outcome suggested that this cytokine might serve as a marker of LN progression, since its serum levels were significantly higher in patients with LN compared with individuals with SLE without kidney involvement. Moreover, a positive correlation between eotaxin and advanced stages of LN was demonstrated.

Similarly, in the present study, we aimed to assess eotaxin as a potential marker of CKD occurrence in a pediatric population. As mentioned above, higher eotaxin serum and urine levels were noted in the study group. Moreover, in order to strengthen those findings, ROC curves were created. In terms of serum eotaxin as a potential marker of CKD occurrence in children, the ROC curve showed 88% sensitivity and 93% specificity. However, there were no significant correlations between eotaxin fluctuations and CKD stage in the presented group.

To date, urine eotaxin has been scarcely analyzed in the literature in terms of kidney diseases. A study, designed by Vieira et al. [33], aimed to assess concentrations of inflammatory markers in male fetuses diagnosed with posterior urethral valve (PUV). Among other interesting findings, the authors indicated a negative correlation between urine eotaxin concentrations and urinary levels of creatinine. In the presented research, urine eotaxin was elevated significantly only in boys with CKD; however, further research on a larger group is required.

Our study is innovative since evaluating eotaxin concentration in children with CKD has not been assessed yet. Nevertheless, one limitation of this study is the that a relatively small group of participants were used; therefore, subsequent analyses on larger cohorts are essential.

## 4. Materials and Methods

### 4.1. Studied Groups

The study group included patients aged 2 to almost 18 years (children and adolescents), diagnosed with chronic kidney disease (*n* = 32) in stages 2 to 5. Children with CKD were divided into groups depending on the disease stage, giving 6 patients in stage 2, 7 patients in stage 3, 10 patients in stage 4 and 9 patients in stage 5. CAKUT, nephronophthisis, FSGS and HUS were listed among factors triggering CKD in children included in the study. The patients were treated in the Department of Pediatric Nephrology with the Subdivision of Dialysis at the Clinical Hospital No. 1 in Zabrze, Medical University of Silesia in Katowice. The following inclusion criteria were applied for this group: confirmed diagnosis of chronic kidney disease in stages 1 to 5—decrease in eGFR for at least 3 consecutive months; pediatric patients—less than 18 years of age. The exclusion criteria included the following: acute kidney injury (AKI), non-malignant and malignant proliferative disorders, infections or inflammatory disorders occurring up to 30 days before sample collection and lack of consent to participate in the study.

The control group (*n* = 41) was represented by patients hospitalized in the Department of Pediatric Nephrology with the Subdivision of Dialysis due to bedwetting or admitted to the Department of Surgery of Child Developmental Defects and Traumatology of the Clinical Hospital No.1 in Zabrze, Medical University of Silesia in Katowice due to one-day surgery procedures. These patients were not diagnosed with chronic diseases nor infectious diseases and their kidney function was normal. Groups of patients were additionally divided into children (aged 2–11 years) and adolescents (12 to almost 18 years). The group of children (2–11 years) with CKD was represented by 19 patients, while the adolescents (11 to almost 18) by 13 participants. Moreover, 30 children and 11 adolescents formed the control group. This research project was approved by the Ethical Committee of the Medical University of Silesia in Katowice (PCN/CBN/0052/KB1/121/22). Written informed consent was obtained from caregivers for all the children and, in the case of participants older than 16 years, also from the child.

### 4.2. Laboratory Tests

Analyzed laboratory tests for the study group included the following: full blood count, urea, creatinine, uric acid, blood ionogram, cholesterol, triglycerides, total serum protein, serum albumin, parathormone (PTH), c-reactive protein (CRP), vitamin D3 concentration, alanine transaminase level (ALT) and transferrin. Revised Schwartz equation for eGFR (0.413 × height [cm]/serum creatinine [mg/dL] [mL/min/1.73 m^2^]) was used for both groups.

### 4.3. Eotaxin Concentration

Eotaxin concentration evaluation in serum and urine was performed with ELISA protocol (Cloud-Clone Corp. (CCC, Houston, TX, USA) kit—Human ECF cat. no. SEA025 Hu). The analytical procedure was completed in accordance with the instructions included by the manufacturer. Absorbance readings were performed using the SYNERGY/H1 reader (BioTek, Santa Clara, CA, USA) at a wavelength of 450 nm, using a reference wave of 570 nm. The results were processed using the Gen5 v 3.05 program (BioTek, Santa Clara, CA, USA). The sensitivity of the method was 5.8 pg/mL. The precision of the method in the simultaneous series (imprecision) was 7.2%.

### 4.4. Anthropometric Measurements

Anthropometric parameters, such as weight, height and blood pressure, were estimated for all study participants. Weight measured in kilograms (with 0.1 kg precision) and height in centimeters (with 0.1 cm precision) by means of a standardized stadiometer were used to calculate Body Mass Index (BMI) (using the equation: weight/height^2^ (kg/m^2^)). The distribution of the above-mentioned parameters was presented on the percentile charts (obtained from OLA and OLAF trials [34,35] and adapted for the population of Polish children). Moreover, standard deviation score (SDS) values for height, weight, BMI, as well as systolic and diastolic blood pressure, were estimated. There was no missing data in the collected groups.

### 4.5. Statistical Analysis

Statistical analysis was performed in the “R studio” program using the “R” programming language. Normality of the distribution of variables was assessed based on the Shapiro–Wilk test. Relationships between variables were assessed using Student's *t*-test for quantitative variables with a distribution close to normal and the Wilcoxon and Kruskal–Wallis tests for quantitative variables with a distribution other than normal, assuming a significance level of *p* < 0.05. Correlation was assessed using the Spearman rank correlation coefficient. Due to numerous interactions between variables, an attempt was made to perform Principal Component Analysis (PCA), but this did not allow for the separation of dimensions that adequately described the studied variables. Single-factor logistic regression models were performed to assess the effect of the studied variables on the membership in the study group. A model that took into account the interaction of the group with the gender of the subjects was also performed. Moreover, ROC curves, plots of the true positive rate (TPR) against the false positive rate (FPR) at each threshold setting, were created for the studied markers. A two-way ANOVA method, used to establish how two independent (categorical) variables influence a dependent (quantitative) variable, assessed the potential impact of the age of participants (children and adolescents) on serum and urine eotaxin concentrations.

## 5. Conclusions

Although the incidence of CKD in the pediatric population is lower than in adults, it still poses a considerable challenge in clinical practice. Therefore, early diagnosis, proper treatment and efficient complications prevention are crucial to achieve good quality of life for children with CKD.

This research proposed a thesis that eotaxin might serve as a marker of CKD occurrence in a pediatric population. Such an outcome has not yet been analyzed in the literature. However, the relatively small group of patients is a particular limitation of this analysis; therefore, further studies on larger cohorts are required.

## Figures and Tables

**Figure 1 ijms-26-07260-f001:**
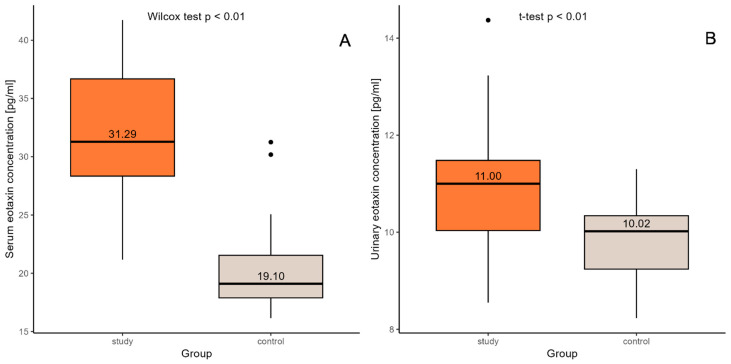
Comparison of eotaxin concentrations in serum (**A**) and urine (**B**) between the control group and the study group. Black dot represents extreme values obtained in the outcome.

**Figure 2 ijms-26-07260-f002:**
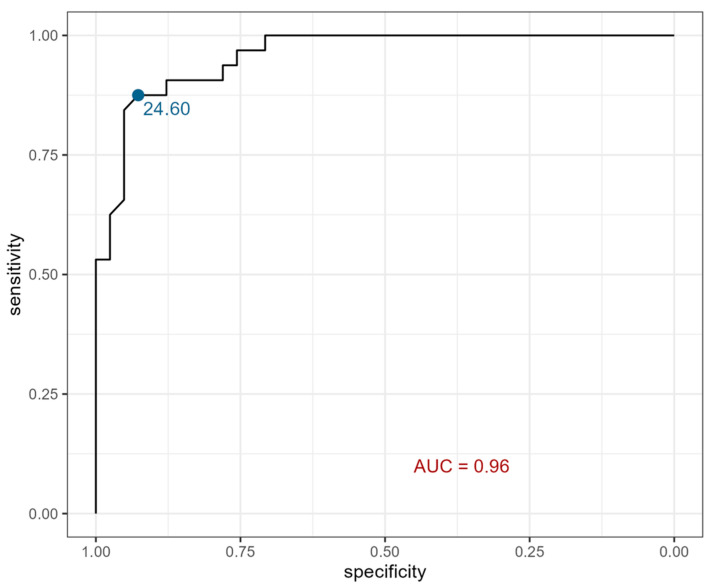
ROC curve for serum eotaxin. AUC—area under the ROC curve; this serves to discriminate individuals as with CKD or as healthy.

**Figure 3 ijms-26-07260-f003:**
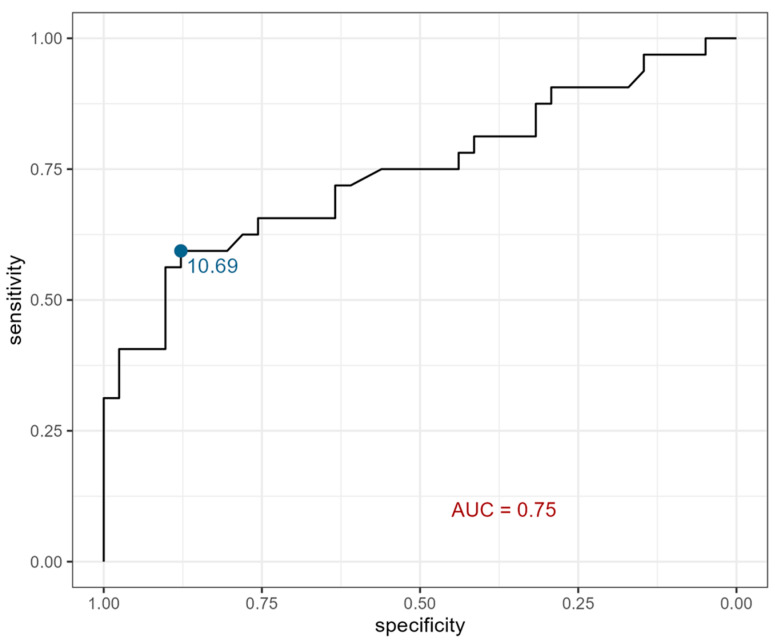
ROC curve for urine eotaxin. AUC—area under the ROC curve; this serves to discriminate individuals as with CKD or as healthy.

**Table 1 ijms-26-07260-t001:** Characteristics of the groups—selected anthropometric and laboratory parameters.

Parameter	Children with Chronic Kidney Disease (*n* = 32)			Control Group (*n* = 41)		
	Whole Group	Female	Male	Whole Group	Female	Male
Age (year)	10.23 ± 5.62 (1.17–17.90)	12.38 ± 4.23 * (6.50–17.90)	9.25 ± 5.98 (1.17–17.90)	9.29 ± 4.19 (2.00–17.50)	8.18 ± 4.01 (2.00–17.00)	10.08 ± 4.21 (4.50–17.50)
Height (cm)	131.86 ± 29.73 (75.00–171.00)	144.30 ± 19.59 * (113.00–165.00)	126.21 ± 32.14 * (75.00–171.00)	136.37 ± 25.28 (82.00–197.00)	125.97 ± 21.23 (82.00–170.00)	143.73 ± 25.72 (110.00–197.00)
SDS for Height	−1.14 ± 1.70 * (−5.69–4.00)	−0.66 ± 0.67 (−1.85–37.29)	−1.36 ± 1.98 (−5.69–4.00)	0.19 ± 1.11 (−1.53–3.00)	−0.17 ± 1.11 (−1.53–2.35)	0.44 ± 1.07 (−1.34–3.00)
Body Weight (kg)	33.13 ± 16.56 (10.00–68.70)	39.06 ± 15.46 (21.70–68.70)	30.44 ± 16.67 * (10.00–65.70)	35.51 ± 19.26 (9.70–87.50)	29.67 ± 15.20 (9.70–71.00)	39.64 ± 21.01 (15.00–87.50)
SDS for Body Weight	−0.72 ± 1.91 * (−3.40–7.33)	−0.51 ± 1.39 (−2.76–2.57)	−0.81 ± 2.13 * (−3.40–7.33)	0.10 ± 1.13 (−2.34–2.12)	0.08 ± 1.27 (−1.87–2.11)	0.11 ± 1.05 (−2.34–2.12)
BMI (kg/m^2^)	17.72 ± 2.93 (14.30–27.20)	18.20 ± 3.76 (15.50–27.20)	17.50 ± 2.55 (14.30–22.70)	17.63 ± 3.53 (12.40–26.50)	17.51 ± 3.21 (13.90–24.60)	17.72 ± 3.81 (12.40–26.50)
SDS for BMI	−0.12 ± 1.59 (−2.47–5.22)	−0.18 ± 1.63 (−2.47–2.85)	−0.10 ± 1.62 * (−2.37–5.22)	−0.01 ± 1.22 (−2.97–2.29)	0.27 ± 1.07 (−1.38–2.29)	−0.22 ± 1.29 (−2.97–2.02)
SYS (mmHg)	106.81 ± 13.58 (83.00–133.00)	106.10 ± 13.54 (88.00–123.00)	107.14 ± 13.91 (83.00–133.00)	111.85 ± 11.35 (85.00–134.00)	107.41 ± 8.73 (89.00–122.00)	115.00 ± 12.08 (85.00–134.00)
DIA (mmHg)	66.38 ± 10.26 (49.00–88.00)	64.10 ± 11.03 (49.00–77.00)	67.41 ± 9.98 * (53.00–88.00)	75.00 ±11.56 (45.00–107.00)	66.82 ± 11.81 (45.00–96.00)	71.38 ± 11.24 (57.00–107.00)
MAP (mmHg)	79.85 ± 10.39 (63.33–103.00)	78.10 ± 11.49 (63.33–91.00)	80.65 ± 10.03 * (67.67–103.00)	83.61 ± 10.33 (62.33–115.70)	80.36 ± 9.83 (62.33–102.33)	85.92 ± 10.25 (70.00–115.70)
Serum creatinine (µmol/L)	210.47 ± 150.21 * (51.00–629.00)	239.80 ± 189.41 * (56.00–629.00)	197.14 ± 131.66 * (51.00–580.00)	49.29 ± 10.03 (32.00–80.00)	45.82 ± 8.88 (32.00–72.00)	51.75 ± 10.25 (39.00–80.00)
eGFR (mL/min/1.73 m^2^)	34.66 ± 22.55 * (9.40–80.90)	38.17 ± 26.77 * (9.40–78.00)	33.06 ± 20.85 * (10.48–80.90)	101.90 ± 12.32 (81.69–138.27)	101.11 ± 10.35 (86.20–117.93)	102.46 ± 13.73 (81.69–138.27)
Serum eotaxin (pg/mL)	31.87± 5.70 * (21.17–41.72)	32.16 ± 6.29 (21.75–41.72)	31.74 ± 5.57 * (21.17–40.54)	20.28± 3.37 (16.15–31.25)	19.84 ± 3.73 (16.15–31.25)	20.58 ± 3.14 (17.03–30.18)
Urine eotaxin (pg/mL)	10.83 ± 1.25 (8.55–14.37)	10.54 ± 1.03 (8.92–12.47)	10.97 ± 1.34 * (8.55–14.37)	9.86± 0.79 (8.23–11.3)	9.71 ± 0.98 (8.23–11.30)	9.97 ± 0.63 (8.79–11.20)

Data are presented as the mean ± standard deviation (minimum–maximum); SDS: standard deviation score; BMI: body mass index; SYS: systolic arterial pressure; DIA: diastolic arterial pressure; MAP: mean arterial pressure; eGFR: estimated glomerular filtration rate. * *p* < 0.05 (study group vs. control group).

**Table 2 ijms-26-07260-t002:** Selected laboratory parameters for the study group.

Parameter	Children with Chronic Kidney Disease		
	Whole Group (*n* = 32)	Female (*n* = 10)	Male (*n* = 22)
Serum urea (mmol/L)	13.47 ± 6.43 (4.10–26.00)	12.19 ± 7.14 (5.20–26.00)	14.05 ± −6.16 (4.10–25.60)
Serum uric acid (umol/L)	371.09 ± 104.52 (120.00–617.00)	375.60 ± 95.63 (234.00–475.00)	369.05 ± −110.41 (120.00–617.00)
Serum total protein (g/L)	68.43 ± 8.25 (43.30–81.30)	69.12 ± 10.67 (43.30–81.30)	68.12 ± −7.16 (47.90–77.00)
Serum albumin (g/L)	44.62 ± 6.21 (18.90–54.00)	43.83 ± 9.64 (18.90–54.00)	44.98 ± −4.08 (34.20–50.38)
Cholesterol (mmol/L)	4.57 ± 1.18 (2.90–8.94)	4.81 ± 1.65 (3.60–8.94)	4.46 ± −0.93 (2.90–6.19)
Triglycerides (mmol/L)	1.61 ± 1.05 (0.52–4.15)	1.38 ± 1.09 (0.52–4.15)	1.71 ± −1.03 (0.59–3.84)
High-density lipoprotein (mmol/L)	1.43 ± 0.39 (0.59–2.20)	1.47 ± 0.35 (0.71–1.80)	1.40 ± −0.42 (0.59–2.20)
Serum hemoglobin (g/dL)	11.82 ± 1.76 (8.60–15.60)	11.77 ± 1.69 (8.60–14.00)	11.85 ± −1.83 (9.70–15.60)
Hematocrit (%)	34.41 ± 5.15 (25.00–46.40)	34.52 ± 4.69 (25.00–41.00)	34.36 ± −5.44 (27.50–46.40)
White blood count (10^3^/uL)	7.22 ± 2.11 (3.96–11.97)	7.60 ± 2.02 (5.00–11.21)	7.05 ± −2.17 (3.96–11.97)
Platelet count (10^3^/uL)	255.03 ± 58.47 (165.00–424.00)	270.80 ± 61.16 (169.00–372.00)	247.86 ± −57.20 (165.00–424.00)
Granulocytes (10^3^/uL)	3.34 ± 1.74 (1.24–8.43)	4.29 ± 1.85 (2.06–8.43)	2.91 ± −1.54 (1.24–7.65)
C-reactive protein (mg/L)	2.51 ± 6.17 (0.01–34.70)	1.59 ± 1.47 (0.42–4.72)	2.93 ± −7.40 (0.01–34.70)
Parathormone (pg/mL)	163.54 ± 187.57 (23.67–835.00)	251.41 ± 291.48 (30.67–835.00)	123.60 ± −100.88 (23.67–337.80)
Serum phosphate (mmol/L)	1.60 ± 0.39 (1.04–2.57)	1.74 ± 0.48 (1.22–2.57)	1.53 ± −0.33 (1.04–2.40)
Serum sodium (mmol/L)	139.00 ± 2.64 (132.00–145.00)	139.10 ± 2.85 (132.00–142.00)	138.95 ± −2.61 (133.00–145.00)
Total serum calcium (mmol/L)	2.46 ± 0.18 (3.20–6.13)	2.42 ± 0.21 (2.05–2.62)	2.47 ± −0.17 (2.06–2.80)
Serum potassium (mmol/L)	4.60 ± 0.67 (3.20–6.13)	4.41 ± 0.57 (3.79–5.70)	4.67 ± −0.70 (3.20–6.13)
D3 vitamin (ng/mL)	37.96 ± 13.77 (7.62–63.84)	34.91 ± 14.39 (7.62–55.00)	39.35 ± −13.59 (13.11–63.84)
Transferrin (umol/L)	25.78 ± 11.50 (9.00–60.00)	32.19 ± 14.04 (11.50–60.00)	22.39 ± −8.53 (9.00–36.60)
Alanine Transaminase (U/L)	18.60 ± 12.04 (6.20–69.70)	12.68 ± 4.39 (6.20–21.10)	21.29 ± −13.47 (8.80–69.70)
Serum bicarbonate (mmol/L)	23.17 ± 3.55 (16.40–30.00)	23.11 ± 3.29 (18.50–28.00)	23.19 ± −3.73 (16.40–30.00)

Data are presented as the mean ± standard deviation (minimum–maximum).

## Data Availability

The original contributions presented in this study are included in the article.

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
