# Peer review of "Assessment of Eotaxin Concentration in Children with Chronic Kidney Disease"

_ijms, 2025, doi:10.3390/ijms26157260_

Round 1

Reviewer 1 Report

Comments and Suggestions for Authors

The authors provide anecdotal evidence that eotaxin is a suitable biochemical marker for CKD in children. Despite some limitations, mainly the rather small size of patient cohorts, briefly mentioned by the authors, the study is well conducted, resulting in reasonable conclusions and is of potential interest to the readership of IJMS.

However, some amendments are recommended:

  1. The interpretation of the ROC curves needs to be described in more detail. In particular, the approach to assign the cutoff points must be specified (e. g. as described in  https://doi.org/10.1155/2017/3762651).
  2.  The sentence l. 131-133 in the Discussion needs to be clarified, as immunomodulation should not be classified as pathology.
  3. Is it justified to include individuals aged 18 years? How is the accepted practice concerning pediatric studies? A more detailed description might be advisable.
Comments on the Quality of English Language

Some editing by a native speaker will be useful.

Author Response

The authors provide anecdotal evidence that eotaxin is a suitable biochemical marker for CKD in children. Despite some limitations, mainly the rather small size of patient cohorts, briefly mentioned by the authors, the study is well conducted, resulting in reasonable conclusions and is of potential interest to the readership of IJMS.

However, some amendments are recommended:

  1. The interpretation of the ROC curves needs to be described in more detail. In particular, the approach to assign the cutoff points must be specified (e. g. as described in  https://doi.org/10.1155/2017/3762651).

Answer: Thank you for this comment. Informations were added.

  1.  The sentence l. 131-133 in the Discussion needs to be clarified, as immunomodulation should not be classified as pathology.

Answer: Thank you for the suggestion. The sentence was altered.

  1. Is it justified to include individuals aged 18 years? How is the accepted practice concerning pediatric studies? A more detailed description might be advisable.

Answer: Thank you for pointing that out. Our patients did not reach 18 years (as mentioned in inclusion criteria), however the oldest participants were past their 17th birthday. The information was corrected.

Reviewer 2 Report

Comments and Suggestions for Authors
  • The manuscript entitled “Assessment of eotaxin concentration in children with chronic kidney disease” assesses the concentration of eotaxin in a group of patients with CKD. It is a well-structured study, but the key limitation is the lack of age-stratified analysis. This study did not separately analyze outcomes by pediatric age subgroups (children vs. adolescents), which may mask age-related variations due to the broad age range (2–18 years) of the patients. According to WHO regulatory definitions, children are from 2 years to 11 years old, and adolescents are from 12 years to 17 or 18 years old. However, the age of the patients in this study ranged from 2 to 18 years old, which means the study includes adolescents, and not all of them are children. Based on this, to strengthen the study’s validity, analyses should compare eotaxin concentrations between children and adolescents, as age-related variations in kidney function, hormonal changes, and other parameters. Authors need to define this point and modify the related parts in the manuscript.
  • Number the subtitles under the title 3 and 4 to be 3.1, 3.2, etc.
  • To improve clarity, some scientific terms should be explained further. For example, chemotactic factor, ROC curve
  • To improve the discussion part, write a short paragraph discussing limitations and weakness points of this study
  • References no. 4, 5, and 6 are outdated. Replace them with updated, well-established references.

Author Response

The manuscript entitled “Assessment of eotaxin concentration in children with chronic kidney disease” assesses the concentration of eotaxin in a group of patients with CKD. It is a well-structured study, but the key limitation is the lack of age-stratified analysis. This study did not separately analyze outcomes by pediatric age subgroups (children vs. adolescents), which may mask age-related variations due to the broad age range (2–18 years) of the patients. According to WHO regulatory definitions, children are from 2 years to 11 years old, and adolescents are from 12 years to 17 or 18 years old. However, the age of the patients in this study ranged from 2 to 18 years old, which means the study includes adolescents, and not all of them are children. Based on this, to strengthen the study’s validity, analyses should compare eotaxin concentrations between children and adolescents, as age-related variations in kidney function, hormonal changes, and other parameters. Authors need to define this point and modify the related parts in the manuscript.

Answer: Thank you for this comment. We added the analysis to the manuscript.

Number the subtitles under the title 3 and 4 to be 3.1, 3.2, etc.

Answer: The subsections were added.

To improve clarity, some scientific terms should be explained further. For example, chemotactic factor, ROC curve

Answer: Thank you for the suggestion. We explained above mentioned terms.

To improve the discussion part, write a short paragraph discussing limitations and weakness points of this study

Answer: Than you for this note. The paragraph was added at the end of discussion.

References no. 4, 5, and 6 are outdated. Replace them with updated, well-established references.

Answer: Thank you for the comment. Reference no. 4 was added to present the research introducing eotaxin for the first time, therefore we believe it could be left in the manuscript. References no. 5 and 6 were replaced.

Reviewer 3 Report

Comments and Suggestions for Authors

This manuscript investigated, according to the abstract and study aim, whether eotaxin, an eosinophil chemoattractant, might serve as a marker of CKD occurrence in a pediatric population.

The introduction needs more references and more carefully worked out background information. Not referenced are statements like the “pathogenesis of CDK in children is different than in adults (line 40), that congenital anomalies of the kidneys and of the urinary tract are frequent causes of CKD in children (line 41/42). Lines 43-45 lists a number of kidney diseases, but does not provide information, whether these conditions also apply to children.

Line 58-62: this sentence contains a lot of information but without the proper references. Please add. Especially a reference supporting a role of eotaxin in the development and/or progression of kidney disease is missing.

Line66/67: the referenced paper shows that eotaxin concentrations may be correlated with impaired kidney development and growth in mice fetuses because of an immune response activation in the mother/dam. Please consider that mouse embryos and mouse pups are different compared to human pediatric patients.

Results:

Undefined abbreviations: ROC, SDS

Line 95/96: the authors write that “serum and urine eotaxin concentrations were markedly higher in the study group, compared to the control group [Fig. 1 A, B], however it did not differ significantly depending on the CKD stage”. My problem is that no separation of the stage of CKD is shown in this manuscript. Based on the GFR the patients were in stages 2-5 (according to the American Kidney Fund).

Since the authors determined the eotaxin concentration in serum and urine, a correlation to CKD should be presented independently of its significance. Another possibility is to use any of the lab parameter (GFR, serum creatinine) to perhaps establish a correlation between stage of CKD and eotaxin.

Line 96/97: The authors state that “Considering outcome for particular gender, urine eotaxin levels were remarkably elevated only in male patients”. Again, data to support this statement are not shown. Please add.

Problem with table 1: the presented values for the whole group are the same as for the male participants. Please correct.

Table 2: Control group is missing.

Figure 2 and 3: please define ROC. Please explain what these traces present with respect to the patients/participants.

Discussion: Please focus the discussion on the provided data and how they support the aim of this study.

Author Response

This manuscript investigated, according to the abstract and study aim, whether eotaxin, an eosinophil chemoattractant, might serve as a marker of CKD occurrence in a pediatric population.

The introduction needs more references and more carefully worked out background information. Not referenced are statements like the “pathogenesis of CDK in children is different than in adults (line 40), that congenital anomalies of the kidneys and of the urinary tract are frequent causes of CKD in children (line 41/42). Lines 43-45 lists a number of kidney diseases, but does not provide information, whether these conditions also apply to children.

Answer: Thank you for the comment. The references were added.

Line 58-62: this sentence contains a lot of information but without the proper references. Please add. Especially a reference supporting a role of eotaxin in the development and/or progression of kidney disease is missing.

            Answer: Thank you for the suggestion. Actually, proper refereces for this sentence are cited in the discussion, therefore we decided to remove this part from introduction (the removed sentence:” The involvement of eotaxin in development, progression and complications of kidney diseases, including CKD, was evaluated in the literature, mainly focusing on correlation with loss of kidney function, increase of serum creatinine and impact on development of renal involvement in systemic diseases, such as type 2 diabetes mellitus (DM2) or systemic lupus erythematosus (SLE). Still, data regarding pediatric population is scarce.”)

Line 66/67: the referenced paper shows that eotaxin concentrations may be correlated with impaired kidney development and growth in mice fetuses because of an immune response activation in the mother/dam. Please consider that mouse embryos and mouse pups are different compared to human pediatric patients.

Answer: Thank you for the comment. The sentence was altered to be more precise. Our intention was to emphasize, that eotaxin might exert an impact on kidney development.

Results:

Undefined abbreviations: ROC, SDS

            Answer: Thank you for the note. The abbreviations were explained.

Line 95/96: the authors write that “serum and urine eotaxin concentrations were markedly higher in the study group, compared to the control group [Fig. 1 A, B], however it did not differ significantly depending on the CKD stage”. My problem is that no separation of the stage of CKD is shown in this manuscript. Based on the GFR the patients were in stages 2-5 (according to the American Kidney Fund).

            Answer: Thank you for the suggestion. The information regarding the numer of patients in each stage was added.

Since the authors determined the eotaxin concentration in serum and urine, a correlation to CKD should be presented independently of its significance. Another possibility is to use any of the lab parameter (GFR, serum creatinine) to perhaps establish a correlation between stage of CKD and eotaxin.

            Answer: Thank you for the suggetion. We added a short information about not significant analyzed correlations.

Line 96/97: The authors state that “Considering outcome for particular gender, urine eotaxin levels were remarkably elevated only in male patients”. Again, data to support this statement are not shown. Please add.

            Answer: Thank you for the suggestion. Data was added to the Table 1.

Problem with table 1: the presented values for the whole group are the same as for the male participants. Please correct.

            Answer: Thank you for the note. We corrected the table.

Table 2: Control group is missing.

            Answer: Thank you for the comment. Unfortunately parameters from the Table 2 were not established for the control group during our study.

Figure 2 and 3: please define ROC. Please explain what these traces present with respect to the patients/participants.

            Answer: Thank you for the suggestion. The ROC curve was explained.

Discussion: Please focus the discussion on the provided data and how they support the aim of this study.

            Answer: Thank you for the suggestion. Since the data regarding eotaxin in kidney diseases is limited we decided to present all available studies on the topic.

Round 2

Reviewer 2 Report

Comments and Suggestions for Authors

The authors have addressed all the comments, and I recommend this manuscript for publication.

Author Response

Thank you very much.

Reviewer 3 Report

Comments and Suggestions for Authors

Please check the value for eotaxin for males in table 1. The given serum concentration of 311.74±5.57 does not fit the measured range of 21.17-40.54.

Author Response

Please check the value for eotaxin for males in table 1. The given serum concentration of 311.74±5.57 does not fit the measured range of 21.17-40.54.

Answer: Thank you for the comment. Indeed, there was a mistake, we corrected the value to 31.74±5.57.